# Evaluation of Tyrosinase Inhibitory Activity of Carbathioamidopyrazoles and Their Potential Application in Cosmetic Products and Melanoma Treatment

**DOI:** 10.3390/ijms26083882

**Published:** 2025-04-19

**Authors:** Ewelina Namiecińska, Jan Jaszczak, Paweł Hikisz, Mateusz Daśko, Magdalena Woźniczka, Elzbieta Budzisz

**Affiliations:** 1Department of the Chemistry of Cosmetic Raw Materials, Medical University of Lodz, Muszynski 1 Str., 90-151 Lodz, Poland; ewelina.namiecinska@umed.lodz.pl; 2Department of Physical and Biocoordination Chemistry, Faculty of Pharmacy, Medical University of Lodz, Muszyńskiego 1, 90-151 Lodz, Poland; jan.jaszczak@umed.lodz.pl (J.J.); magdalena.wozniczka@umed.lodz.pl (M.W.); 3Department of Oncobiology and Epigenetics, Faculty of Biology and Environmental Protection, University of Lodz, Pomorska 141/143, 90-236 Lodz, Poland; pawel.hikisz@biol.uni.lodz.pl; 4Department of Inorganic Chemistry, Faculty of Chemistry, Gdańsk University of Technology, Narutowicza 11/12, 80-233 Gdansk, Poland; mateusz.dasko@pg.edu.pl

**Keywords:** tyrosinase, carbothioamidopyrazole derivatives, melanin, hyperpigmentation, molecular docking

## Abstract

Hyperpigmentation can be prevented by regulating melanin synthesis through tyrosinase inhibition. As such, tyrosinase inhibitors like arbutin, kojic acid, and hydroquinone are commonly used for skin lightening. Recent studies suggest that certain pyrazole derivatives with tyrosinase activity may also have anticancer potential by influencing melanocyte transformation and tumor progression, positioning them as promising candidates for both cosmetic and therapeutic uses. The aim of this study was to evaluate the tyrosinase inhibitory activity of carbothioamidopyrazole derivatives. Inhibition was determined using the Dixon method, leveraging in silico molecular docking and circular dichroism (CD) spectroscopy to analyze fluorescence quenching. Carbothioamidopyrazole derivatives at the C-3 and C-5 positions in the pyrazole ring may be effective alternatives to traditional skin-lightening agents. These derivatives can induce structural changes in tyrosinase, thus altering its activity, and influence melanocyte transformation. Their dual action as tyrosinase inhibitors and potential anticancer agents makes them valuable for future research. Two compounds exhibited stronger inhibitory activity than kojic acid. Molecular docking suggests that these derivatives may block tyrosinase activity by preventing substrate access to its active site. These results underscore the potential of pyrazole derivatives for both cosmetic and therapeutic applications.

## 1. Introduction

Tyrosinase is a key enzyme responsible for melanin biosynthesis in the skin, making it a significant target for both cosmetic and anticancer therapies, particularly those targeting melanoma. This enzyme is widely distributed in fungi, higher plants, and animals and is involved in the first two steps of melanin biosynthesis, where it is responsible for the hydroxylation of tyrosine to the left-handed form of the amino acid dopamine (*L*-DOPA) and the oxidative conversion of *L*-DOPA to dopachinone [1,2]. This enzymatic activity not only contributes to skin pigmentation but also influences the behavior of melanoma cells, making tyrosinase a dual target in therapeutic strategies.

In cosmetic applications, tyrosinase inhibition is a fundamental strategy for treating hyperpigmentation disorders such as melasma and age spots [1]. Pigmentation is responsible for the color of the skin, eyes, and hair and protects the tissues under the skin from harmful UV radiation. UV rays accelerate the aging process of the skin or lead to carcinogenesis, and ultimately melanoma [3]. While melanin has a number of valuable protective properties, its overproduction can cause aesthetic problems such as melasma, freckles and even age-related lentigines [4]. As such, there is great interest in identifying agents that can correct any excess or inadequacy of pigment in the skin by influencing the tyrosinase responsible for pigment biosynthesis. The search for skin bleaching agents has also become an important target for the cosmetic industry, where a large number of moderate to potent tyrosinase inhibitors of both natural and synthetic origin have been reported over the past decade [5,6,7,8,9]. For example, kojic acid, hydroquinone, and arbutin are widely used in skin lightening products. These agents act by directly inhibiting the enzymatic activity of tyrosinase, thereby reducing melanin synthesis and promoting a more even skin tone [10,11]. In clinical studies conducted to date, only arbutin and kojic acid have been found to inhibit pigmentation production in intact melanocytes [12], while hydroquinones are considered cytotoxic to melanocytes and potentially mutagenic to mammalian cells [11]. Kojic acid is also believed to cause contact dermatitis, sensitization, redness, and erythema [13].

Furthermore, the exploration of natural compounds as tyrosinase inhibitors has gained popularity in both the cosmetic and therapeutic fields. Extracts from various plants have been identified as potential sources of new tyrosinase inhibitors that may prove to be safer alternatives to synthetic agents [14]. The use of natural products not only fits into the growing consumer demand for clean and sustainable cosmetic products, but may also incur fewer side effects than traditional synthetic inhibitors [15]. For example, *Glycyrrhiza glabra* (licorice) root extract contains glabridin, which effectively inhibits melanogenesis [16], while *Morus alba* (white mulberry) root bark and *Camellia sinensis* (green tea) extracts are known for their content of oxyresveratrol and catechins, respectively, both showing notable anti-tyrosinase activity [17,18]. Additionally, a variety of other plant-derived compounds, including phenolic acids and flavonoids such as apigenin, luteolin, and quercetin, as well as their glycosides, have demonstrated promising tyrosinase inhibitory properties [19].

Targeting tyrosinase may also be an approach to treating melanoma, a malignant tumor of melanocytes characterized by excessive melanin production. Melanoma cells typically express elevated levels of tyrosinase, which not only contributes to skin pigmentation but also plays a role in cell survival and proliferation. As such, targeting tyrosinase presents a dual opportunity: it can potentially reduce melanin production, thereby addressing cosmetic concerns, while also impairing the growth of melanoma tumors [20,21]. Indeed, studies have shown that tyrosinase inhibition can lead to reduced melanin synthesis, reduced tumor growth and metastasis in melanoma models [22,23].

The therapeutic potential of tyrosinase inhibitors is further underscored by their ability to sensitize melanoma cells to other treatment modalities. By reducing melanin levels, these inhibitors can enhance the efficacy of chemotherapeutic agents, allowing for lower doses and reduced side effects [24]. This synergistic effect is particularly important in the context of melanoma, where resistance to conventional therapies presents a significant challenge; as such, the ongoing development of new inhibitors, combined with a deeper understanding of the molecular mechanisms involved, offers hope for improving therapeutic outcomes in melanoma patients [25,26].

Synthetic tyrosinase inhibitors have become a common approach for addressing skin pigmentation issues, yet many of these compounds are associated with undesirable side effects, highlighting the need for new, more effective alternatives. Among the most promising candidates are carbathioamidopyrazoles, a group of compounds that not only can exhibit strong bleaching effects but also can hold potential for protecting against melanoma, one of the most prevalent and aggressive forms of skin cancer. Beyond their ability to inhibit tyrosinase activity, these compounds have demonstrated impressive anticancer properties [9,10,11], making them potential dual-purpose agents in both cosmetic and therapeutic applications. What sets carbathioamidopyrazoles apart is their broad spectrum of biological activities, including anticancer, antifungal, antimicrobial, and antimalarial effects [27]. Furthermore, the distinctive structure of this group of compounds, including the presence of a characteristic thione moiety containing a sulfur atom capable of binding to the dicopper center of tyrosinase—as previously demonstrated by Ghani and Ullah in their study—has served as an inspiration for undertaking this line of research [28,29]. These versatile properties make them highly attractive for a variety of therapeutic purposes, offering potential applications far beyond the realm of dermatology. Given their multifaceted potential, these compounds represent an exciting avenue for further investigation in both clinical and cosmetic fields.

The aim of this study was to evaluate the activity of carbothioamidopyrazole derivatives with various substituents (methyl, ethyl, hydroxyl, phenyl) in the C-3 and C-5 position of the pyrazole ring as tyrosinase inhibitors and to identify their mode of activity (Figure 1).

## 2. Results

### 2.1. Molecular Docking to Tyrosinase

Initially, to verify that the newly-designed compounds **1**–**4** are able to effectively bind to the tyrosinase active site, molecular docking studies were performed. The X-ray structures of tyrosinase was retrieved from the Protein Data Bank (code: 2Y9X) and prepared for docking calculations. The docking procedure of the optimized ligands was performed using AutoDock Vina 1.1.2 software (Molecular Graphics Laboratory, The Scripps Research Institute, La Jolla, CA, USA). The calculated results indicated that compounds **1**–**4** could, at least theoretically, efficiently bind to the active site of the enzyme. The free binding energies of compounds **1**–**4** were satisfactory, as summarized in Table 1. Compounds **1, 2**, and **4** demonstrated free energy of binding values in the narrow range of −5.2 to −5.4 kcal/mol, similar to those calculated for the reference kojic acid (−5.7 kcal/mol). Only compound **3** demonstrated significantly higher free energy of binding (−2.1 kcal/mol), indicating lower binding abilities.

The docking experiment showed that compounds **1**–**4** may bind to tyrosinase in a very similar manner to kojic acid (Table 2). They may occupy the same region of the tyrosinase active site, thus interacting with many catalytically crucial amino acid residues, including numerous histidine residues present in the catalytic site. For example, compound **4**, which demonstrated the highest binding ability, was found to dock a short distance from Gly281, His263, and His259 and copper ions. This close proximity may suggest the presence of interactions that may stabilize the compound–enzyme complex. The binding modes for the other examined derivatives are shown in Table 2.

### 2.2. The Dixon Kinetics Test

The kinetic mechanisms of inhibition were determined for carbothioamidepyrazole derivatives **1**–**4** and kojic acid using the Dixon kinetic test [30], with the latter being used as a reference. The experiments used the following Formula (1):1/*v* = 1/*V*(1 + *K_m_*/*s*)(1 + *i*/*K_i_*), (1)
where:*v*—initial rate of inhibition reaction;*V*—maximum velocity obtained at high substrate concentrations;*K_m_*—Michaelis constant;*K_i_*—Inhibition constant;*i*—concentration of inhibitor;*s*—concentration of substrate.

The obtained results as shown Figure 1, Figure 2, Figure 3, Figure 4 and Figure 5. Tyrosinase, which contains copper ions in its active site, becomes more active when exposed to UV light. Kojic acid binds to copper ions and prevents tyrosinase activation. By inhibiting tyrosinase activity, kojic acid can also prevent melanin formation [8].

The molecular docking analysis found that the newly synthesized compounds block the access of the tyrosinase enzyme to the active site. Compounds **1** and **4** show greater inhibitory capacity than the reference kojic acid (Table 3), and hence may demonstrate competitive inhibition. For competitive inhibitors, a few lines at a few different substrate concentrations for a series of inhibitor concentrations intersect in the second quadrant at the value of –Ki [31].

### 2.3. Changes in Tryptophan Fluorescence Intensity–Fluorescence Quenching Mechanism

Measurements of tryptophan fluorescence intensity changes in proteins serve as a valuable tool in biophysical studies, enabling the monitoring of protein conformational changes and interactions with other molecules. Any alteration in protein conformation that affects the position of the tryptophan residue relative to other groups, or its solvent accessibility, can alter its fluorescence intensity and lifetime. The binding of different molecules to the protein can influence tryptophan fluorescence by inducing conformational changes.

To gain a deeper understanding of the mechanism underlying tyrosinase inhibition by the investigated compounds, the impact of carbathioamidopyrazole derivatives on the intrinsic fluorescence of tyrosinase was examined. Tryptophan residues are primarily responsible for the fluorescence spectrum of tyrosinase, i.e., at approximately 340 nm following excitation at 280 nm. It can be seen that tyrosinase fluoresced at 334 nm upon excitation at 280 nm (Figure 6), while the tested derivatives exhibited negligible fluorescence under identical conditions.

A 24 h incubation of tyrosinase with the tested derivatives led to a substantial decrease in the intensity of tryptophan fluorescence. This effect was apparent for all four compounds, although with compounds **1** and **4** quenching tryptophan fluorescence to a greater degree than **2** and **3**.

This quenching of fluorescence intensity suggests that the carbathioamidopyrazole derivatives can interact with tyrosinase, thus altering its conformation. Interestingly, in addition to the clear decrease in fluorescence intensity, all test compounds demonstrated a slight shift of the fluorescence intensity peak towards longer wavelengths, by about ~10 nm (red shift in the emission wavelength). This result may indicate that tryptophan residues were transferred to a more hydrophilic microenvironment after interacting with the tested compounds, which could cause conformational changes of the enzyme. The fact that the carbathioamidopyrazole derivatives mask the tryptophan of tyrosinase implies that the tyrosinase may became disagglomerated and that its structure was loosened.

Similarly to our present findings, previous research has also demonstrated that interactions between pyrazole derivatives and tyrosinase can induce substantial alterations in the microenvironment surrounding tryptophan residues [32,33]. The binding of such derivatives was found to modify the hydrophobicity and polarity of the tryptophan environment, consequently influencing its fluorescent properties. Fluorescence quenching can also arise from the formation of ground-state complexes, as indicted by fluorescence spectroscopy studies, which found the intrinsic fluorescence intensity of tyrosinase to decrease in the presence of pyrazole derivatives [34].

### 2.4. Changes in the Secondary Structure of Tyrosinase

The secondary structure and conformational dynamics of tyrosinase can be elucidated by circular dichroism (CD) spectroscopy. The CD spectrum of tyrosinase is typically characterized by negative bands centered around 209 nm and 221 nm, which are indicative of π–π* and n–π* electronic transitions associated with α-helical structural elements; a deeper minima is associated with a greater share of helices in the protein structure. A positive maximum around 195 nm may indicate the presence of beta-sheets. In turn, the lack of clear characteristic features may indicate a large share of disordered structures (loops, regions without a defined structure) [35].

As shown in Figure 7, the control tyrosinase solution exhibited characteristic peaks in its CD spectrum: a positive peak around 196 nm and two negative peaks at approximately 208 nm and 221 nm. The CD spectra of tyrosinase incubated with the tested carbothioamidopyrazole derivatives clearly demonstrate interactions between these compounds and the protein, leading to alterations in its peptide structure.

Compounds **1** and **4** induced the most significant changes in tyrosinase secondary structure. Slightly smaller conformational changes were observed after incubation with compound **2**. Notably, 24 h incubation with these three derivatives consistently resulted in greater conformational changes compared to one-hour incubation. Interestingly, the CD spectra of tyrosinase incubated with compound **3** remained unchanged regardless of incubation time.

Overall, CD spectroscopy revealed that the tested carbothioamidopyrazole derivatives can destabilize the tyrosinase structure. Analysis of the CD spectra indicated that compounds **1**, **2**, and **4** induce specific conformational changes in tyrosinase’s secondary structure. A consistent trend was observed for all three derivatives, with the most pronounced changes occurring at wavelengths associated with the α-helical form of the protein. Incubation with these derivatives led to a decrease in intensity at the 208 nm and 221 nm peaks. Additionally, compound **1** induced a slight decrease in the intensity of the 196 nm peak, potentially indicating structural changes within *β*-sheets.

Based on the results of the CD analysis, the percentage changes in α-Helix, β-Turn, and random coil were estimated using CD Spectra Deconvolution software (Figure 8). This software is a deconvolution analysis tool that can be used to analyze CD data in the far UV spectral region to obtain more detailed quantitative analysis of secondary structure. Deconvolution was performed in the spectral region 195–260 nm due to its accuracy in this region (i.e., with the total percentage sum closest to 100%). The obtained results indicate that compounds **1** and **4** caused significant changes in the tyrosinase structure, primarily in the *α*-Helix structure: a percentage decrease in *α*-Helix was associated with a simultaneous increase in *β*-Turn and random coil structures.

## 3. Discussion

Tyrosinase plays a key role in melanogenesis and skin browning. As such, it has become a focus for numerous cosmetic studies aimed at identifying effective inhibitors with bleaching properties [36]. In an oncological context, however, it has been proposed that tyrosinase may reduce the proliferation of abnormal melanocytes [37].

Tyrosinase inhibitor activity is typically evaluated by measuring dopachrome formation using substrates such as tyrosine (monophenolic) or *L*-DOPA (diphenolic) [38]. In the present study, the kinetic mechanisms of inhibition were determined through the application of the Dixon method, a commonly-employed technique for the analysis of competitive and mixed-type inhibitors.

The obtained results indicate that the tested compounds bind to the enzyme’s active site in a competitive manner, i.e., by chelating copper ions in the active center, as is the case with kojic acid, thus effectively inhibiting melanin biosynthesis [39]. This result is in agreement with those of our molecular docking study, which found that compounds **1**–**4** can bind to tyrosinase in a very similar manner to kojic acid; indeed, the free energy of binding calculated for compound 4 (−5.4 kcal/mol) was similar to that of kojic acid (−5.7 kcal/mol). The compounds hence appear to occupy the same region of the tyrosinase active site, where they interact with numerous key amino acid residues, including the histidine residues present in the catalytic site.

The carbathioamidopyrazole derivatives containing a methyl group in the C-3 position of the pyrazole ring (compounds **1** and **4**) demonstrated the greatest inhibitory properties against tyrosinase. Compound **4** showed twice the effect of kojic acid. It hence appears that the presence of both groups together, i.e., the methyl group in the C-3 position and the hydroxyl group in the C-5 position of the pyrazole ring, is beneficial for inhibition of tyrosinase. However, the presence of an arenyl substituent instead of an alkyl substituent at the C-3 position significantly impairs its action, suggesting that substitution at the C-3 and C-5 positions is essential for blocking the enzyme. The presence of alkyl substituents in both these positions shows moderate blocking. Hence, the studied carbathioamidopyrazole derivatives appear to effectively block tyrosinase.

In our previous studies, the derivatives were found to exhibit noteworthy antioxidant properties and showed no cytotoxicity toward normal HFF-1 fibroblasts or human microvascular endothelial cells (HMEC-1) [7,27,40]. This antioxidant property may help mitigate oxidative stress, which is known to contribute to DNA damage and the neoplastic transformation of melanocytes, thereby potentially offering a protective effect [41]. These compounds not only contribute to aesthetic depigmentation, but may also reduce the proliferation of abnormal melanocytes. Moreover, the observed lack of cytotoxicity toward normal cells may suggest a favorable safety profile, providing a basis for further research on these compounds as promising candidates for development in both clinical dermatology and cosmetology.

Tyrosinase, a key enzyme in melanin biosynthesis, plays a significant role in melanoma development and progression. This enzyme catalyzes the oxidation of phenolic compounds, which serves as a critical step in melanin production. The relationship between tyrosinase activity and melanoma cell viability is well documented; studies indicate a correlation between increased tyrosinase activity, increased melanin content, and enhanced melanoma cell survival [42]. Mutations in the tyrosinase gene can lead to increased enzyme activity and melanin overproduction, consequently increasing the risk of melanoma development. Furthermore, dysfunctional tyrosinase can result in the formation of abnormal melanin, which can damage DNA and elevate the risk of melanoma-inducing mutations.

The enzymatic activity of tyrosinase extends beyond melanin production, also affecting the redox balance and oxidative stress within melanoma cells. It also catalyzes the conversion of monophenols to diphenols, and subsequently to o-quinones, which can induce the generation of reactive oxygen species (ROS) [43]. Elevated ROS levels can induce oxidative stress, potentially contributing to DNA damage and promoting tumorigenesis. Conversely, the accumulation of tyrosinase-mediated melanin may provide a protective effect against oxidative damage, thereby increasing melanoma cell survival under oxidative stress [44]. This complex interplay between tyrosinase activity, melanin production, and oxidative stress underscores the multifaceted role of this enzyme in melanoma pathophysiology.

In light of the crucial biological role of tyrosinase, the identification of potent inhibitors is of significant therapeutic interest. In our study, compounds **1**–**4** demonstrated the strongest inhibitory potential, as reflected by their low Ki values compared to other tested derivatives. Since Ki is independent of substrate concentration and directly reflects the binding affinity between the inhibitor and the enzyme, it provides an objective and reliable measure of inhibitory potency. Notably, compound **4** exhibited a Ki value approximately two times lower than that of kojic acid, as a widely used reference inhibitor. This result suggests that compound **4** binds more tightly to tyrosinase and may exhibit stronger inhibitory activity under comparable conditions [30].

Tyrosinase is an increasingly promising target for melanoma treatment. Such approaches could exploit the fact that the enzyme is overexpressed in melanoma cells, thus enabling selective targeting of malignant cells while sparing healthy tissue. This specificity could enhance treatment efficacy and minimize side effects, establishing tyrosinase as a valuable target in the fight against melanoma [37]. Furthermore, research into the role of tyrosinase in the immune response to melanoma may lead to the development of more effective immunotherapy strategies.

Tyrosinase inhibition therapy with both natural compounds and synthetic therapeutics have yielded promising effects in melanoma cells. Notably, the use of enzyme inhibitors often results in melanoma suppression, reducing invasiveness and proliferation [20,45,46,47].

Pyrazole derivatives are promising candidates for tyrosinase inhibition, particularly in melanoma anticancer therapy. Their ability to inhibit tyrosinase activity, coupled with potential anticancer properties, makes them valuable candidates for further research and development [48,49,50]. Recent studies, such as those by Tarasiuk et al. [51], emphasize the importance of structural modifications for increasing the inhibitory potential of pyrazole derivatives. The presence of functional groups, such as NH or free NH_2_ groups, is considered crucial in the inhibition mechanism, as they compete with the substrate for binding to the copper active site of tyrosinase.

In addition to direct tyrosinase inhibition, pyrazole derivatives can also modulate the signaling pathways involved in melanogenesis. Iervasi et al. [52] demonstrated that imidazo-pyrazole derivatives affect the proteomic profile of SKMEL-28 melanoma cells, suggesting that these compounds may alter cell signaling to enhance their anti-tumor effects. This dual action, i.e., targeting both the enzyme and signaling pathways, may provide a more comprehensive approach to melanoma treatment.

Research has shown that the binding of small molecules, including pyrazoline derivatives, can lead to significant alterations in the secondary structure of proteins [53]. This suggests that similar interactions may occur with pyrazoline derivatives, potentially leading to a loss of helical structure or other secondary structural elements. Liu et al. [54] investigated the interaction between kojic acid and tyrosinase, providing a valuable model for understanding the impact of small molecules on enzyme conformation; the data indicated that binding by kojic acid induced alterations in the protein backbone, leading to extended polypeptide chains and global conformational changes [54]. These observations suggest that pyrazoline derivatives may similarly influence the secondary structure of tyrosinase, potentially through competitive binding or allosteric mechanisms.

Certain inhibitors can inhibit tyrosinase by disrupting its secondary structure and inducing conformational changes. Research has demonstrated that specific compounds can decrease the α-helix content of tyrosinase while increasing the *β*-sheet and random coil content. These alterations in secondary structure may signify a conformational change that hinders the enzyme’s catalytic activity and compromises its structural integrity [33,55].

Available studies suggest that pyrazoline derivatives may interact with tyrosinase, leading to alterations in its secondary structure, impacting its catalytic efficiency and stability. Vaezi et al. [56] demonstrated that ligand binding to fungal tyrosinase resulted in a slight reduction in α-helical content, suggesting a shift towards a less stable conformation. Our findings also indicate that pyrazoline derivatives may induce conformational rearrangements in tyrosinase, thereby affecting its enzymatic activity.

The observation of fluorescence quenching and a redshift of the emission maximum upon addition of the compound to the protein solution strongly suggests that the compound interacts with the protein structure in proximity to tryptophan residues. A redshift in the tryptophan fluorescence spectrum typically indicates an increase in the polarity of the tryptophan residue environment. In native proteins, tryptophan residues are frequently located within the hydrophobic interior. Interaction of a protein with a given compound can induce conformational changes, leading to partial unfolding and the exposure of tryptophan residues to a more polar solvent [57]. Tyrosinase fluorescence is sensitive to alterations in its environment, influenced by both intrinsic factors, such as protein structure, and extrinsic factors, including the presence of inhibitors or other interacting molecules. Cui et al. [58] demonstrated that specific substituents, such as those in p-substituted cinnamic acid derivatives, can create a polar microenvironment, resulting in significant redshifts in tyrosinase fluorescence spectra.

The observed redshift of the fluorescence maximum of tyrosinase incubated with the tested derivatives is corroborated by the existing literature. As indicated by You et al. [59], the introduction of specific functional groups can influence the binding interactions between compounds and copper ions in the active site of tyrosinase; this mechanism is essential in determining the effectiveness of these compounds in inhibiting melanin production. Spectral changes in fluorescence often signify modifications to the electronic properties of molecules, which can directly affect their inhibitory activity [23]. Hydrophobic and ionic interactions between carbathiamidopyrazoles and tyrosinase induce changes in the microenvironment of residues, leading to observable alterations in fluorescence spectra–specifically, a redshift. This shift is typically associated with a lower energy state upon excitation, indicating that the surrounding environment of the fluorophore (in this case, tryptophan and tyrosine residues in tyrosinase) has become more polar or has been altered due to inhibitor binding [32]. Consistent with Pei’s findings [60], similar mechanisms are observed in protein interactions with small molecules, where the redshift occurs as a result of the exposure of polar residues to a more hydrophilic environment, accompanying unfolding or conformational changes in protein structure. The redshift of fluorescence observed in carbathiamidopyrazole interactions with tyrosinase is a product of complex molecular dynamics influenced by structural conformations, local microenvironmental shifts, and solvent interactions. These phenomena not only reflect the binding stability and inhibitory potency of these compounds but also offer important implications for the design of novel fluorescent probes and therapeutic agents targeting melanin synthesis.

Another study examined the impact of caffeine on the secondary structure of tyrosinase using circular dichroism (CD) spectroscopy; the findings revealed significant alterations in the α-helical content of the enzyme upon caffeine addition [61]. While this study specifically focused on caffeine, it underscores the potential of CD spectroscopy to assess the structural effects of small molecules, including pyrazoline derivatives, on tyrosinase. The structural characteristics of pyrazoline derivatives, particularly their hydrogen-bonding and π–π stacking capabilities, may significantly influence their binding affinity to tyrosinase and thus affect its conformational dynamics [62]. The impact of pyrazoline derivatives on the secondary structure of tyrosinase may be attributed to their inherent chemical properties: the presence of electron-withdrawing or electron-donating groups on the pyrazoline ring can influence the binding affinity and dynamics of its interaction with tyrosinase, with varying inhibitory effects [62]. These findings suggest that the specific chemical structure of pyrazoline derivatives plays a critical role in determining their influence on the secondary structure.

Although this study provides valuable information on the interactions of carbathioamidopyrazole derivatives with tyrosinase and their effects on the enzyme’s structure, it is important to acknowledge certain limitations that may help to refine the interpretation of the results.

First, the study focused on in vitro experiments using purified tyrosinase. While this approach allows for controlled conditions and the isolation of specific interactions, it does not fully replicate the complex cellular environment in which tyrosinase operates. Factors such as cellular metabolism, protein-protein interactions, and the presence of other biomolecules may influence the observed interactions and effects of the tested compounds.

Circular dichroism spectroscopy analysis, while informative regarding secondary structure changes, provides limited insight into the precise binding sites and the nature of the interactions between the derivatives and tyrosinase.

Finally, the study primarily focuses on the direct interaction of the derivatives with tyrosinase. Although this is crucial for understanding the mechanism of inhibition, it does not fully address the potential downstream effects of tyrosinase inhibition on melanogenesis and melanoma cell behavior. Further studies investigating the effects of these derivatives on melanin production, cell proliferation, and other relevant cellular processes would provide a more comprehensive understanding of their therapeutic potential.

In summary, although this study makes a significant contribution to our understanding of the interactions between carbathioamidopyrazole derivatives and tyrosinase, recognizing these limitations is crucial for a balanced interpretation of the results. Addressing these limitations in future studies will undoubtedly enhance the relevance of these findings to the development of melanoma therapies.

Moreover, given that some of the tested carbothioamidopyrazole derivatives—particularly compounds **1** and **4**—demonstrated stronger tyrosinase inhibition than kojic acid, these compounds could be considered promising candidates for use as depigmenting agents in the cosmetic industry, particularly in the development of novel and more effective skin-lightening formulations. Kojic acid, while widely used, has been associated with several drawbacks, including poor photostability, susceptibility to oxidation, and the potential to cause contact dermatitis or allergic reactions in sensitive individuals [63,64].

A wide range of natural and synthetic compounds have been studied for their potential as tyrosinase inhibitors, especially in the context of skin depigmentation. Among the most commonly used agents in cosmetic formulations are kojic acid, arbutin, and azelaic acid [65,66]. Several studies have shown that some inhibitors effective against mushroom tyrosinase display limited activity toward the human enzyme [66]. In contrast, newly developed compounds such as thiamidol and deoxyarbutin have shown higher specificity for human tyrosinase and better skin penetration properties [66,67]. Additionally, α-arbutin has been identified as a more potent alternative to arbutin for reducing melanin production [68]. However, their clinical efficacy, particularly against human tyrosinase, remains controversial.

The promising activity of the investigated carbothioamidopyrazole derivatives highlights their potential to serve as alternative or complementary active ingredients in topical formulations intended for the treatment of hyperpigmentation disorders such as melasma, lentigines (age spots), post-inflammatory hyperpigmentation, and general uneven skin tone. These conditions are among the most common cosmetic concerns and are typically managed using tyrosinase inhibitors [7,68,69]. Notably, many phenolic-based natural compounds, including flavonoids such as quercetin, apigenin, and their glycosides, act via copper-chelating mechanisms at the tyrosinase active site [70]. The results obtained in this study suggest that the tested carbothioamidopyrazole derivatives may act through a similar mechanism, supporting their potential utility in cosmetic applications targeting pigmentary disorders.

In dermatology, the therapeutic potential of bleaching substances extends beyond their depigmenting effects to encompass their ability to modulate the inflammatory pathways involved in skin conditions such as melasma, hyperpigmentation, and age spots [71]. Recent studies have highlighted the importance of targeting the molecular mechanisms of melanogenesis while maintaining skin integrity and preventing inflammatory responses that could lead to adverse effects such as contact dermatitis [72]. Furthermore, compounds with strong tyrosinase inhibition and minimal cytotoxicity may serve as a safer alternative to currently used agents, such as hydroquinone, which has been associated with skin irritation and ochronosis in prolonged use [21]. Therefore, these preliminary studies hold promise and may pave the way for the future application of these compounds in both preventive and therapeutic dermatological treatments.

## 4. Experimental Section, Materials and Methods

All compounds (**1**–**4**) were prepared as described previously [27,73,74,75].

### 4.1. Computational Studies

#### 4.1.1. Ligand Preparation

The 3D structure of the ligands was prepared with the Portable HyperChem 8.0.7 Release (Hypercube, Inc., Gainesville, FL, USA). Prior to docking calculations, the structure of each ligand was optimized using a MM+ force field and the Polak–Ribière conjugate gradient algorithm (terminating at a gradient of 0.05 kcal mol^−1^ Å^−1^).

#### 4.1.2. Protein Preparation

The X-ray structures of the tyrosinase used for molecular modeling studies was taken from the Protein Databank (Protein Data Bank accession code: 2Y9X). After standard preparation procedure (including removal of water molecules, other ligands and amino acid chains B-H, and the addition of hydrogen atoms and Gasteiger charges to each atom) docking analysis was carried out.

#### 4.1.3. Molecular Docking

Docking studies were carried out using Autodock Vina 1.1.2 software (The Molecular Graphic Laboratory, The Scripps Research Institute, La Jolla, CA, USA) [76] with default parameters (exhaustiveness = 8, num_modes = 30, and energy_range = 10). For the docking studies, a grid box size of 12 Å × 12 Å × 12 Å centered on Cγ of His263 residue (x = −7.461 y = −29.768, z = −40.188), which is one of the amino acid residues forming the catalytic pocket of Tyrosinase [77]. Graphic visualizations of the 3D model for the poses with the lowest free energies of binding were generated using VMD 1.9 software (University of Illinois at Urbana-Champaign, Urbana, IL, USA).

### 4.2. The Kinetics Method of Dixon

The kojic acid, tyrosinase mushrooms (EC 1.14.18.1), and 3,4-dihydroxy-*L*-phenylalanine were purchased from Sigma-Aldrich. Dimethyl sulfoxide was purchased from Chempur. Calcium dihydrogen phosphate and disodium phosphate were products of POCH Gliwice. All other reagents were local and analytical grade. The water used was distilled and ion-free. The measurements used a phosphate buffer pH 20 = 6.88. The newly synthesized compounds were first dissolved in DMSO and then in a buffer solution pH20 = 6.88. The final concentration of DMSO was 10%. Before spectrophotometric measurement, the inhibitors were incubated with the tyrosinase enzyme solution at 20 °C for 20 min and the reaction was initiated by adding the *L*-DOPA substrate. Measurements were performed for two minutes from the start of the reaction at a wavelength of 485 nm using a VICTOR X Multilabel Plate Reader (Perkin Elmer, Waltham, MA, USA). Each measurement was repeated three times.

### 4.3. Determination of the Secondary Structure of Tyrosinase—Circular Dichroism (CD)

In order to determine the changes in the secondary structure of tyrosinase under the influence of the investigated carbathioamidopyrazole derivatives, circular dichroism (CD) analysis was performed.

Circular dichroism spectroscopy was detected on a CD spectropolarimeter (Jasco J-815 CD spectrometer, Jasco International Co., Ltd., Tokyo, Japan) with a 1 mm quartz cuvette at room temperature, at a wavelength set from 195 to 260 nm. Stock solutions of investigated compounds at a final concentration of 2 μM were prepared in the phosphate buffer. CD changes were measured one hour after adding the tested compounds to the tyrosinase solution (1h) or after 24 h of incubation (24 h). The final concentration of carbathioamidopyrazole derivatives was 2 µM in the measuring cuvette.

### 4.4. Fluorescence Quenching Mechanism of Tyrosinase

In order to determine the degree of tryptophan fluorescence quenching in tyrosinase, fluorescence intensity was analyzed using a Perkin-Elmer LS-5B spectrofluorometer (Perkin-Elmer LS-5B spectrofluorometer, Waltham, MA, USA). The quenching of fluorescence emitted by tryptophan of tyrosinase upon carbathioamidopyrazole derivative addition was observed at the wavelength emission range of 300–400 nm (λ_ex_ 280 nm). The monochromator slits were set at 5 nm for excitation and 5 nm for emission. The final concentration of the tested derivatives in the measuring cuvette was 2 µM, and the concentration of tyrosinase was 1 µg/mL. The enzyme was incubated with the tested derivatives for 24 h, after which the fluorescence intensity was measured.

## 5. Conclusions

The pyrazole derivatives studied here may represent promising candidates for potential use in both depigmenting cosmetic products and in future anticancer therapies. Our results suggest that that carbathioamidopyrazole derivatives containing a methyl group in the C-3 position of the pyrazole ring (compounds **1** and **4**) were the best inhibitors of tyrosinase, with compound 4 exhibiting much better effects than kojic acid. Our study further indicates that all tested compounds competitively bind to tyrosinase at its active site, as confirmed by Dixon’s test and molecular docking calculations. Additionally, other research has suggested that carbathioamidopyrazole derivatives may induce conformational changes in tyrosinase, potentially affecting its enzymatic activity. These effects could have relevance in the neoplastic transformation of melanocytes, a topic that warrants further exploration.

Hence, the ability of the tested compounds to inhibit tyrosinase activity, combined with their potential anticancer properties, make them valuable candidates for further research and development. However, the challenge will be to synthesize compounds that strike a balance between reducing excess pigmentation and protecting the skin from harmful environmental factors.

Due to their ability to inhibit tyrosinase, these compounds demonstrate potential as active ingredients in cosmetic products aimed at reducing skin discoloration and evening out skin tone. Further studies on their stability and efficacy in cosmetic formulations are necessary. Future studies should focus on the synthesis and characterization of new derivatives of these compounds, optimizing their structure to increase the efficacy and selectivity of tyrosinase inhibition. Further in-depth also studies of the molecular mechanisms by which carbathioamidopyrazoles inhibit tyrosinase activity are necessary. The use of spectroscopic techniques and molecular modeling can provide valuable information on the interactions of these compounds with the enzyme. Prior to potential use in cosmetic products or anticancer therapy, it is crucial to conduct comprehensive in vitro and in vivo studies assessing the safety and toxicity of carbathioamidopyrazoles and their derivatives on normal cells.

In conclusion, this article provides a basis for further interdisciplinary research on carbathioamidopyrazoles, which may lead to the development of new effective and safe solutions in the field of cosmetology and potentially in melanoma therapy.

## Data Availability

Data are contained within the article.

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
