# Peer review of "Evaluation of Tyrosinase Inhibitory Activity of Carbathioamidopyrazoles and Their Potential Application in Cosmetic Products and Melanoma Treatment"

_ijms, 2025, doi:10.3390/ijms26083882_

Round 1
Reviewer 1 Report
Comments and Suggestions for Authors
The manuscript entitled “Evaluation of Tyrosinase Inhibitory Activity of Carbothioamidopyrazoles and Their Potential Application in Cosmetic Products and Melanoma Treatment”, is relatively well-written and fits within the journal's scope; however, the authors need to make some minor changes.
General points:
- The manuscript is well-organized and highlighted the significance of the topic. It covers the standard sections: the introduction, methods, important results, followed by a conclusion.
- The methods are well-designed and presented clearly, making perfect sense. The combination of different techniques to substantiate the study’s claims strengthens the validity of the results.
- The findings provide new perspective into alternative tyrosinase inhibitors that may be more effective and safer than existing agents. The compounds' dual cosmetic and therapeutic relevance emphasized the study's significance. However, I strongly suggest to maintain uniformed boldness and resolution for the chemical structures in Table.1. I understand that compounds 1 and 4 were intentionally emphasized because they were the best inhibitors of the tyrosinase enzyme, a more visually uniformed presentation of all compounds would be preferable. You may further highlight these compounds by enclosing them in a circle or box for emphasis.
- The authors presented an outstanding discussion to make the manuscript interesting for readers, though the discussion should explicitly acknowledge any limitations in the study to strengthen the interpretation of the findings.
- The study is relevant to both the cosmetic and pharmaceutical industries; however, it would strengthen the manuscript if further discussion is included on how these findings could be translated into industry applications.
- The conclusion connected with the title, however; future directions and clinical applications are lacking in this section.
Specific points:
- The study's molecular docking details are satisfactory, but the rationale for choosing specific docking parameters could be better justified. Also, the molecular docking images should include binding site interactions more explicitly.
- The Dixon kinetic analysis description should clarify how competitive inhibition was determined (beyond the observed Ki values).
- Further discussion for the observed red shift in the fluorescence quenching might be beneficial.
- Figures and data display are well-labeled; yet, the resolution of some graphs (especially Dixon plots) could be enhanced.
- Although the references are thorough, the over-reliance on specific sources should be noticed and justified. Authors may consider diversifying the citations with additional recent references. Moreover, self-citations appear to be within reasonable limits and are relevant to the discussion. Still, there is a noticeable dependance on references from the authors' previous publications. Although these citations may be appropriate, the authors should consider balancing them with additional independent studies to increase the manuscript's credibility.
- The presented data strongly supports the conclusion, though future research directions such as safety assessments and potential clinical applications, should be briefly noted.
Overall recommendation: Reconsider after minor revisions
Although the manuscript is of high quality and presents significant findings, certain elements of data presentation and methodological explanation can be implemented and require revision.
Comments on the Quality of English LanguageOverall, the quality of the English language is satisfactory; however, minor refinements could improve readability.
Author Response
The response to the Reviewers Comment
Response to the Reviewer 1
Comment 1: The findings provide new perspective into alternative tyrosinase inhibitors that may be more effective and safer than existing agents. The compounds' dual cosmetic and therapeutic relevance emphasized the study's significance. However, I strongly suggest to maintain uniformed boldness and resolution for the chemical structures in Table.1. I understand that compounds 1 and 4 were intentionally emphasized because they were the best inhibitors of the tyrosinase enzyme, a more visually uniformed presentation of all compounds would be preferable. You may further highlight these compounds by enclosing them in a circle or box for emphasis.
Response 1: Thank you for this comment. Of course, we fully agree with the reviewer's suggestion. We have followed the reviewer's advice and have standardized the boldness and resolution of all chemical structures in Table 1 to ensure a more uniform presentation. We believe these changes improve the clarity and consistency of the table.
Comment 2: The authors presented an outstanding discussion to make the manuscript interesting for readers, though the discussion should explicitly acknowledge any limitations in the study to strengthen the interpretation of the findings.
Response 2: We appreciate your comment. As suggested, we have added a paragraph to the discussion regarding some limitations of our study, and have indicated the direction of future experiments aimed at deepening our understanding of the molecular basis of tyrosinase inhibition by carbathioamidopyrazoles.
Comment 3: The study is relevant to both the cosmetic and pharmaceutical industries; however, it would strengthen the manuscript if further discussion is included on how these findings could be translated into industry applications.
Response 3: Thank you for this insightful suggestion. We have incorporated additional discussion in the manuscript on the potential industry applications of our findings, highlighting their relevance to both the cosmetic and pharmaceutical sectors. We believe this addition strengthens the manuscript and provides a clearer perspective on the practical implications of our study.
Comment 4: The conclusion connected with the title, however; future directions and clinical applications are lacking in this section.
Response 4: Thank you for this comment. We have revised the conclusion to include future directions and potential clinical applications, ensuring aligns with the study's objectives. We believe these additions enhance the clarity and impact of our conclusions.
Comment 5: The study's molecular docking details are satisfactory, but the rationale for choosing specific docking parameters could be better justified. Also, the molecular docking images should include binding site interactions more explicitly.
Response 5: Thank you for this attention. Regarding the docking parameters, we used the default parameters of the Autodock Vina software and we did not need to change them - we added this information in the revised manuscript. For docking studies we chose Tyrosinase model 2Y9X retrieved from the Protein Databank according to the protocol reported in the literature [Chen J.; Ye Y.; Ran M.; Li Q.; Ruan Z.; Jin N., Frontiers in Pharmacology, 2020, 11, 81]. The grid box was centered on Cγ of His263 residue, which is one of the amino acid residues forming the catalytic pocket of Tyrosinase as indicated in the same reference (you can find this information in the manuscript). Regarding images, we tried to improve them, however, due to the quite large number of interactions it was hard to make all of them more visible in the one figure. Therefore, we decided to add additional column in the Table 2, where we listed the distances between compounds and amino acid residues in each case.
Comment 6: The Dixon kinetic analysis description should clarify how competitive inhibition was determined (beyond the observed Ki values).
Response 6: Thank you for this comment. We have added this information to the manuscript.
Comment 7: Further discussion for the observed red shift in the fluorescence quenching might be beneficial.
Response 7: Following the reviewer's request, we have added an appropriate paragraph to the discussion concerning the mechanisms responsible for the quenching of tyrosinase fluorescence and, above all, the shift of the fluorescence emission maximum towards longer wavelengths (fluorescence red shift).
Comment 8: Figures and data display are well-labeled; yet, the resolution of some graphs (especially Dixon plots) could be enhanced.
Response 8: Thank you for your comment. The resolution of the graphs, particularly the Dixon plots, has been improved to ensure better clarity and visibility.
Comment 9: Although the references are thorough, the over-reliance on specific sources should be noticed and justified. Authors may consider diversifying the citations with additional recent references. Moreover, self-citations appear to be within reasonable limits and are relevant to the discussion. Still, there is a noticeable dependance on references from the authors' previous publications. Although these citations may be appropriate, the authors should consider balancing them with additional independent studies to increase the manuscript's credibility.
Response 9: Thank you for this comment. We have added additional independent studies to increase the manuscript's credibility.
Comment 10: The presented data strongly supports the conclusion, though future research directions such as safety assessments and potential clinical applications, should be briefly noted.
Response 10: Thank you for this comment. We have added a relevant paragraph to the conclusion. We hope that it will meet the Reviewer's expectations.
Reviewer 2 Report
Comments and Suggestions for Authors
The activity of tyrosinase enzymes influences the functioning of melanoma cells. Accordingly, tyrosinase inhibitors play a pivotal role in cosmetics as a prevention and/or inhibition of hyperpigmentation and are also trying their beneficial effects in tumor therapy. So far, the effects of several enzyme inhibitors have been tested to inhibit tyrosinase, but they may have side effects. The authors investigated the effect of four carbothioamidopyrazole derivatives on mushroom tyrosinase, mainly by monitoring the changes in the primary and secondary structures of the enzyme. The effects of the four derivatives were compared with kojic acid, a known tyrosinase inhibitor. Derivatives 1 and 4 (their chemical names are not given in the article, but their structures are shown) showed a greater inhibitory effect than the reference kojic acid. These compounds also showed a statistically significant change in the structure of the tyrosinase enzyme, primarily in the α-Helix structure, and considering the secondary structure, in the simultaneously increased β-Turn and random coil structure.
The article presents a very thorough literature review and discussion of the results, using precision methods. The results obtained are fully acceptable and we are eagerly awaiting the results of the effects of individual derivatives on tumor cells in cell lines that have not yet been studied in the literature.
It is worth noting that it contains no typos and the English is completely adequate and fine..
My comments and questions :
1, Why is it that when testing the substrate L-dopa in the Dixon plots, at a substrate concentration of about 800 micromolar and inhibitor concentrations of 0.5 and 1 mM, no results can be seen for compounds 3 and 4 (Fig. 3 and Fig.4)?
2, Hydroquinone, a tyrosinase inhibitor has a carcinogenic effect. Can such an effect be reported for the applied derivatives, despite the fact that the control of tumor cells is one of the goals of their application?
3, In Scheme 1, I recommend providing the exact chemical names of the four derivatives, because it makes the content and essence of the article easier to follow for some readers (mainly chemists).
Author Response
Response to Reviewer 2
Comment 1: Why is it that when testing the substrate L-dopa in the Dixon plots, at a substrate concentration of about 800 micromolar and inhibitor concentrations of 0.5 and 1 mM, no results can be seen for compounds 3 and 4 (Fig. 3 and Fig.4)?
Response 1: Thank you for your comment. In the case of compounds 3 and 4, at low inhibitor concentrations (0.5 and 1 mM) and high substrate concentration (~800 µM L-DOPA), it was not possible to obtain consistent and reliable data due to the high experimental error in this range. The signal-to-noise ratio was too low to allow meaningful interpretation of the results. For this reason, the corresponding data points are not shown in Figures 3 and 4.
Comment 2: Hydroquinone, a tyrosinase inhibitor has a carcinogenic effect. Can such an effect be reported for the applied derivatives, despite the fact that the control of tumor cells is one of the goals of their application?
Response 2: Thank you for your comment. We have added information according to the suggestion on future research directions, including safety assessments and potential clinical applications, in the manuscript.
Comment 3: In Scheme 1, I recommend providing the exact chemical names of the four derivatives, because it makes the content and essence of the article easier to follow for some readers (mainly chemists).
Response 3: Thank you for this comment. We have added the exact chemical names on scheme 1 of the four derivatives according to the suggestion.
Reviewer 3 Report
Comments and Suggestions for Authors
In the work titled "Evaluation of tyrosinase inhibitory activity of carbathioamidopyrazoles and their potential application in cosmetic products and melanoma treatment" authors analyze synthetic tyrosinase inhibitors as alternative for kojic acid and arbutin.
My sugestions:
Introduction:
line 43: citation is missing
There are many tyrosinase inhibitiors derived from plants, please ad proper examples - extracts and single compounds; with citations
Please explain more clearly why you have chosen this group of compounds : lines 93-105
Results:
Table 1- molecular skeletons are barely visible (2,3, kojic acid)
Molecular docking- Do the tested compounds only reach the active site or do they also show affinity for other structural elements of the enzyme?
Discussion: lines from 289- 307 I would like a more objective explanation of why the authors believe that compounds 1-4 are the best candidates for tyrosinase inhibition. I do not quite understand from the results from which point it follows that compound 4 blocks tyrosinase twice as strongly as kojic acid. IC50 has not been calculated
Have any tests been performed on melanoma lines to confirm that the compounds tested will effectively inhibit cancer development?
in my opinion, such optimistic conclusions lack extensive in vitro and in vivo studies. Please reconsider your conclusions
Author Response
Response to Reviewer 3
Comment 1: Introduction: line 43: citation is missing.
Response 1: Thank you for this comment. We have added the citation in line 43. The citation is: Tyrosinase Inhibitors from Natural and Synthetic Sources as Skin-lightening Agents; Mohammad N. Masum, Kosei Yamauchi, Tohru Mitsunaga; Reviews in Agricultural Science 7:41-58; https://doi.org/10.7831/ras.7.41.
Comment 2: Introduction: There are many tyrosinase inhibitiors derived from plants, please ad proper examples - extracts and single compounds; with citations.
Response 2: Thank you for this comment. We have added appropriate examples of plant-derived tyrosinase inhibitors, including both extracts and single compounds, along with relevant citations.
Comment 3: Introduction: Please explain more clearly why you have chosen this group of compounds : lines 93-105.
Response 3: Thank you for this comment. We have provided a clearer explanation in the manuscript regarding the selection of this group of compounds.
Comment 4: Results: Table 1- molecular skeletons are barely visible (2,3, kojic acid).
Molecular docking- Do the tested compounds only reach the active site or do they also show affinity for other structural elements of the enzyme?
Response 4: Thank you for this comment. Generally, the docking studies are performed for specific regions of molecular target that may be responsible for the binding sites of ligands. The docking studies of ligands using the structure of whole molecular target are more complicated and much more time-consuming. In our case, we examined only the affinity of the compounds to the active site of Tyrosinase using the grid box centered on Cγ of His263 residue, which is one of the amino acid residues forming the catalytic pocket as indicated in the literature [Chen J.; Ye Y.; Ran M.; Li Q.; Ruan Z.; Jin N., Frontiers in Pharmacology, 2020, 11, 81]. Therefore, we can not exclude that they demonstrate affinity for the other structural elements of the molecular target.
Comment 5: Discussion: lines from 289- 307 I would like a more objective explanation of why the authors believe that compounds 1-4 are the best candidates for tyrosinase inhibition. I do not quite understand from the results from which point it follows that compound 4 blocks tyrosinase twice as strongly as kojic acid. IC50 has not been calculated.
Response 5: Thank you for this valuable comment. We agree that ICâ‚…â‚€ values provide a useful comparative measure, but in our study we focused on the determination of inhibition constants (Káµ¢), which are considered more reliable for mechanistic interpretation, as they are independent of substrate concentration. The Káµ¢ values for compounds 1–4 were significantly lower than for the other tested compounds, which suggests a stronger binding affinity to tyrosinase. In particular, compound 4 exhibited a Káµ¢ value approximately two times lower than that of kojic acid, indicating a higher inhibitory potency under the tested conditions.
We have clarified this point in the Discussion section (lines 289–307) to ensure a more objective and transparent justification of why compounds 1–4 were highlighted as the most promising inhibitors in our study.
To substantiate this information and enhance the credibility of the statement, we have added a relevant literature [d] citation in the revised manuscript.
[31]Burlingham B.T.; Wilanski T.S. An Intuitive Look at the Relationship of Ki and IC50: A More General Use for the Dixon Plot Concepts in Biochemistry 2003, 80(2), 214-218
Comment 6: Have any tests been performed on melanoma lines to confirm that the compounds tested will effectively inhibit cancer development?
Response 6: In 2023, an article was published (https://doi.org/10.3390/molecules28093969) where we investigated the molecular basis of the anticancer activity of Arene–Ruthenium(II) Complexes with Carbothioamidopyrazoles. In the studies, we used a number of different human cancer cell lines, including tumorigenic primary melanoma cell line WM115. Some of the tested derivatives showed very attractive cytotoxic and anticancer properties against the mentioned cell line.
Comment 7: In my opinion, such optimistic conclusions lack extensive in vitro and in vivo studies. Please reconsider your conclusions.
Response 7: Thank you for this comment. We have revised the conclusions to reflect a more cautious and balanced perspective. While the in vitro results are promising, we acknowledge the need for further extensive in vitro and in vivo studies to fully validate the potential of the compounds. We have revised the conclusions to emphasize that additional research, including in vivo experiments, is necessary before making definitive claims regarding their efficacy and safety.